# IL-17RA promotes pathologic epithelial inflammation in a mouse model of upper respiratory influenza infection

Zahrasadat Navaeiseddighi[1ʘ], Jitendra Kumar Tripathi[2,3ʘ], Kai Guo[4ʘ], Zhihan Wang[2,5ʘ], Taylor Schmit[2], Delano R. Brooks[1], Reese A. Allen[1], Junguk Hur[2], Ramkumar Mathur[3], Donald Jurivich[3], Nadeem Khan[1,2]*

1 Dept of Oral Biology, College of Dentistry, University of Florida, Gainesville, Florida, United States of America, 2 Department of Biomedical Sciences, School of Medicine and Health Sciences, University of North Dakota, Grand Forks, North Dakota, United States of America, 3 Department of Geriatrics, School of Medicine and Health Sciences, University of North Dakota, Grand Forks, North Dakota, United States of America, 4 Department of Neurology, University of Michigan, Ann Arbor, Michigan, United States of America, 5 West China School of Basic Medical Sciences & Forensic Medicine, Sichuan University, Chengdu, Sichuan, China

ʘ These authors contributed equally to this work.
* nkhan2@dental.ufl.edu

**Data Availability Statement:** Data generated or analyzed during this study are within the manuscript. RNA sequencing data has been

## Abstract

The upper respiratory tract (nasopharynx or NP) is the first site of influenza replication, allowing the virus to disseminate to the lower respiratory tract or promoting community transmission. The host response in the NP regulates an intricate balance between viral control and tissue pathology. The hyper-inflammatory responses promote epithelial injury, allowing for increased viral dissemination and susceptibility to secondary bacterial infections. However, the pathologic contributors to influenza upper respiratory tissue pathology are incompletely understood. In this study, we investigated the role of interleukin IL-17 recetor A (IL-17RA) as a modulator of influenza host response and inflammation in the upper respiratory tract. We used a combined experimental approach involving IL-17RA[-/-] mice and an air-liquid interface (ALI) epithelial culture model to investigate the role of IL-17 response in epithelial inflammation, barrier function, and tissue pathology. Our data show that IL-17RA[-/-] mice exhibited significantly reduced neutrophilia, epithelial injury, and viral load. The reduced NP inflammation and epithelial injury in IL-17RA[-/-] mice correlated with increased resistance against co-infection by *Streptococcus pneumoniae (Spn)*. IL-17A treatment, while potentiating the apoptosis of IAV-infected epithelial cells, caused bystander cell death and disrupted the barrier function in ALI epithelial model, supporting the *in vivo* findings.

## Author summary

The nasopharynx is the first site of influenza virus replication. The defects in the orchestration or regulation of host response in the upper respiratory tract cause localized tissue damage that can allow the development of a high viral load, resulting in community

deposited into the NCBI Gene Expression Omnibus with accession ID: GSE226679.

**Funding:** This work was supported by the National Institute of Health awards R01AI143741 and R21 AI151522 (to N.K.), the National Institute of Centers of Biomedical Research Excellence award 5P20GM113123 and IDeA Network of Biomedical Research Excellence award 5P20GM103442 (to the University of North Dakota Flow Cytometry Core), and the National Institute of General Medical Sciences awards P20GM113123 (to the University of North Dakota Computational Data Analysis Core), and P20GM113123 and U54GM128729 (to the University of North Dakota Histology core). ZN was supported by R01AI143741. The funders had no role in study design, data collection and analysis, decision to publish, or preparation of the manuscript.

**Competing interests:** All authors have read and approved the final version of the manuscript. The authors have declared that no conflict of interest exists.

transmission or dissemination into the lungs. Further, since the nasopharynx is a primary site of *Streptococcus pneumoniae* bacterial colonization, influenza-induced tissue damage can convert commensal bacterial colonization into an active infection by promoting bacterial outgrowth and dissemination. The mechanisms underlying influenza-induced tissue damage in the nasopharynx are not fully understood. A better understanding of how the influenza virus develops pathologic host response in the nasopharynx could help develop new treatments against influenza and influenza-associated severe bacterial infections by *Streptococcus pneumoniae*. Using an experimental mouse model with localized nasopharyngeal influenza infection, we show that IL-17RA, a receptor for interleukin IL-17A, is a significant contributor to influenza tissue damage in the nasopharynx and promotes severe bacterial co-infection by *Streptococcus pneumoniae*. We show that IL-17RA mediated tissue damage in the nasopharynx results from enhanced recruitment of neutrophils, as well as a direct tissue damage associated function of IL-17RA on nasopharyngeal epithelial cells.

## Introduction

Influenza virus replicates in the upper respiratory tract and can cause a mild localized infection to severe viral pneumonia requiring hospitalization [1,2]. Additionally, influenza host response can convert commensal bacterial colonization into a severe disease by promoting tissue damage and bacterial outgrowth [3,4]. Therefore, a better understanding of the host response to influenza in the upper respiratory tract could help inform the development of new therapies or interventions to prevent or treat influenza.

Epithelial cells are the first site of influenza replication in the NP, which orchestrate a local immune response to execute viral control [5,6]. However, host response-driven viral clearance involves significant bystander epithelial injury that can promote viral spread and make inflamed epithelial cells permissive to enhanced bacterial colonization. The unrestrained viral or bacterial burden in the NP disseminates to sterile organs, i.e., middle ear, sinuses, or lungs, to establish the disease or transmit it into the community. Despite significant published work, there remains a knowledge gap on the pathologic contributors of influenza host response in the upper respiratory tract.

The IL-17RA signaling is crucial to host defense against mucosal pathogens [7]. The protective role of IL-17-IL17RA axis has been widely reported against respiratory bacterial colonization [8,9]. However, the findings from lung models suggest the pathologic contribution of IL-17RA to influenza lung injury [10,11]. Along similar lines, we reported the double-edged role of IL-17A; while protective against *Streptococcus pneumonia* (*Spn*) colonization in NP, Influenza A Virus (IAV) triggered a robust IL-17A response that converted asymptomatic bacterial colonization into an invasive infection [12]. Typically, IL-17-mediated pathogen control at mucosal surfaces involves the regulation of neutrophilic inflammation [13,14] and barrier function [15,16], which guards epithelial cells against pathogen invasion. While the findings from gut inflammatory models suggest the dual role of IL-17A on epithelial barrier function [17–19], there remains a knowledge gap on IL-17-mediated regulation of epithelial inflammation and barrier integrity in the upper respiratory tract during influenza. Because influenza pathogenesis commences in the upper respiratory tract, a better understanding of the nasopharyngeal host response to influenza is desirable.

In this study, we used a combined experimental approach involving IL-17RA$^{-/-}$ mice and an air-liquid interface (ALI) model with human nasopharyngeal epithelial cells (HNEpCs) to

investigate the role of IL-17A/IL-17RA axis in epithelial inflammation and tissue pathology during IAV. Our findings show that IL-17RA$^{-/-}$ mice had reduced neutrophilia and inflammatory response in nasal lumen, which correlated with ameliorated epithelial injury and enhanced resistance against *Spn* co-infection. In ALI model, IL-17A treatment triggered robust epithelial inflammation, resulting in significant apoptosis of viral-infected and bystander cell death, disruption of barrier integrity, and enhanced viral spread, supporting the *in vivo* findings. Our findings provide strong evidence that IL-17RA signaling is a significant pathologic contributor to IAV tissue pathology in the NP, promoting viral spread and bacterial super-infection.

## Results

### IL-17RA regulates neutrophil recruitment in nasopharynx without affecting blood release

While IL-17RA has been shown to regulate neutrophil recruitment in IAV lung models, there is a lack of information on IL-17RA and cellular immune responses in the upper respiratory tract during IAV infection. To investigate the role of IL-17RA as a regulator of inflammation in NP, we intranasally (i.n.) infected WT and IL-17RA$^{-/-}$ (*hereafter KO*) mice with a low volume (10 μL) inoculum of IAV and measured the immune cell frequencies in NP lavage 7 days after IAV infection (7 dpi). Compared to WT, KO mice had significantly reduced neutrophil levels in NP lavage (Fig 1A and 1B), which correlated with reduced CXCL1 levels (Fig 1C). No change in non-neutrophil immune cells (macrophages and monocytes) was observed between IAV-infected WT and KO mice (Fig 1D). To investigate if reduced neutrophil recruitment in KO mice resulted from impaired bone marrow (BM) release into the bloodstream, we compared the blood neutrophil levels between IAV-infected WT and KO mice. Unlike NP lavage, WT and KO mice had comparable levels of neutrophils in the circulation, suggesting that IL-17RA deficiency does not affect neutrophil release into the blood (Fig 1E). Next, we compared the protein levels of inflammatory and Th17 polarizing cytokines (IL-6, TGF-β) in the NP lavage of IAV-infected WT and KO mice. The KO mice did not exhibit defects in IL-17A production (Fig 1F). Compared to WT, KO mice had reduced IL-6, but no difference in TNF-α levels was observed between the two groups (Fig 1F). In contrast, KO mice had increased TGF-β levels (Fig 1F). Overall, these data suggest reduced neutrophilia and inflammatory response in KO mice during IAV upper respiratory infection.

### IL-17RA$^{-/-}$ mice exhibit reduced inflammation, epithelial injury, and viral load in the NP

Because KO mice had reduced neutrophil recruitment, CXCL1, and higher TGF-β levels in the NP lavage, we compared NP tissue pathology between IAV-infected WT and KO mice. The H&E staining of nasal septa was performed to determine tissue inflammation and pathology between the two groups (WT vs. KO) on 3, 7, and 14 days post IAV infection (dpi) time points. Our data show that IAV induced mild (non-signficant) inflammation at 3 dpi, and no difference in nasal tissue inflammation was observed between IAV-infected WT and KO mice on 3 dpi. However, Compared to WT, KO mice displayed significantly reduced inflammation based on leukocytic infiltration (Fig 2A and 2B) at 7 dpi. In addition, IAV-infected WT group exhibited characteristics consistent with damage to the epithelium, including epithelial swelling and ciliary denudation. These are consistent with observations made in prior reports [20]. Changes to the epithelial layer and cilia were less pronounced in IAV-infected KO mice. More specifically, cilia remained largely intact in KO mice, and the thickening of the basement

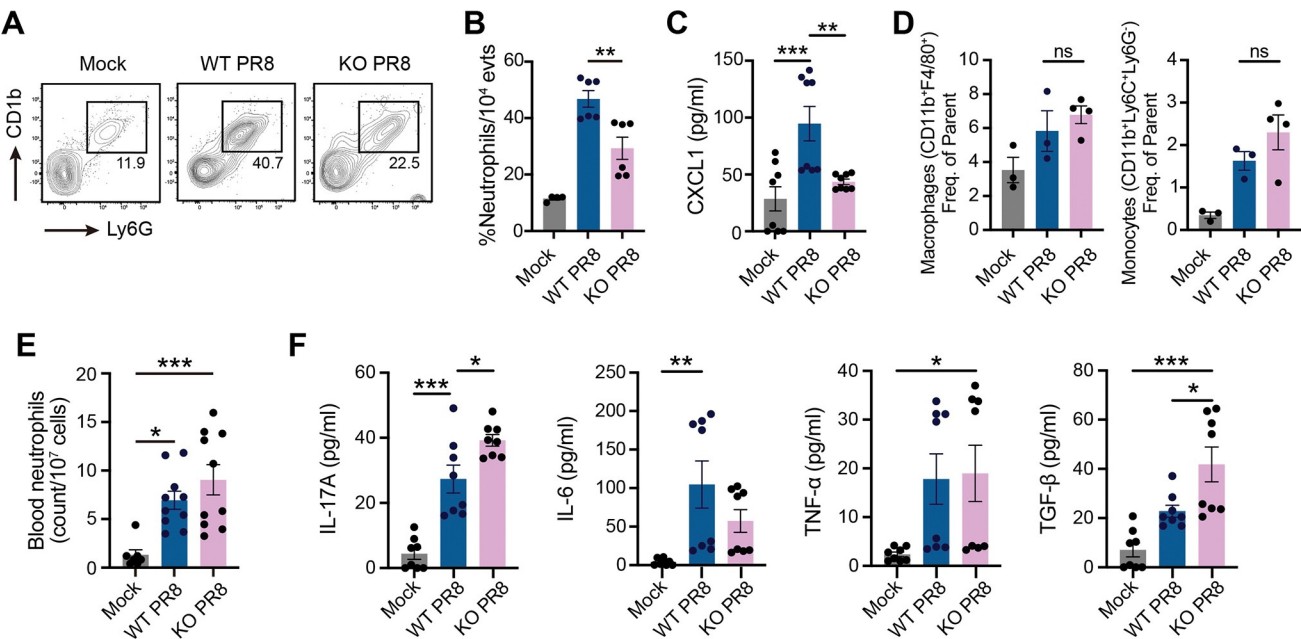

**Fig 1. IL-17RA regulates neutrophilic response via CXCL1 in the NP.** WT C57BL/6 and IL17RA$^{-/-}$ (KO) mice were infected with 10 μl of PR8 (50xTCID$_{50}$) or PBS on day 0. Mice were euthanized at 7 dpi. for cell phenotyping and cytokine analysis. **A-B.** Neutrophils (CD11b$^+$Ly6C$^+$Ly6G$^+$) present in nasal lavage and expressed as % of CD11b$^+$ cells. **C.** CXCL1 chemokine levels in nasal lavage. **D.** Macrophages (CD11b$^+$F4/80$^+$) and monocytes (CD11b$^+$Ly6C$^+$Ly6G$^-$) counts in nasal lavage. **E.** Neutrophils (CD11b$^+$Ly6G$^+$) counts in blood. **F.** Cytokine levels in nasal lavage determined by flow cytometric bead array. Data are representative of two independent experiments (n = 3-4/experiment) and expressed as SEM. For lavage cell phenotyping and cytokine analysis, samples were pooled and split into triplicate. Data were analyzed by One-Way ANOVA with Tukey's post-hoc. *p<0.05, **p<0.01, ***p<0.001; ns, not significant.

membrane was not as pronounced as in the IAV-infected WT group. However, despite the notable differences (between WT and KO) at 7 dpi, both groups exhibited normalized tissue inflammation by 14 dpi and no difference was found between WT and KO mice. Since epithelial injury is a significant risk for viral spread, we measured IAV load in NP lavage of WT and KO mice at 3, 7, and 14 dpi. WT mice (compared to KO) had a higher viral load in the NP lavage on 3 and 7 dpi (Fig 2C). However, no detectable viral load was found in the nasal lavage of WT and KO mice by 14 dpi. These data suggest that IL-17RA induced pathologic effect is largely limited during the peak inflammation (7 dpi), and the presence of IL-17RA doesn't produce long-term defects in the resolution of inflammation or IAV viral control in WT mice.

Next, we purified NP epithelial cells (CD45$^-$ Epcam$^+$) and performed RNAseq to investigate molecular changes in epithelial cells between IAV-infected WT and KO mice. The principal component analysis (PCA) plot revealed distinct expression profiles between mock and IAV infection groups (Fig 3A). The Venn diagram showed that only 11 differentially expressed genes (DEGs) with adjusted p value <0.05 and |log2 fold change| >1 were identified among all group comparisons (Fig 3B). Compared to mock, a total of 358 and 313 DEGs were identified in IAV-infected WT (WT PR8 vs. WT Mock) and KO (KO PR8 vs. KO Mock) mice, respectively. Of those, 84 DEGs were determined under both comparisons (WT PR8 vs. WT Mock and KO PR8 vs. KO Mock), indicating these DEGs were commonly expressed by IAV infection. We performed gene set enrichment analysis (GSEA) for KEGG to identify molecular pathways associated with the activation of immune response (WT PR8 vs. WT Mock, KO PR8 vs. KO Mock, KO PR8 vs. Mock PR8) (Fig 3C). A majority of pathways belonging to the immune system/inflammation, such as the Toll-like receptor signaling pathway, RIG-I-like receptor signaling pathway, NOD-like receptor signaling pathway, and Neutrophil

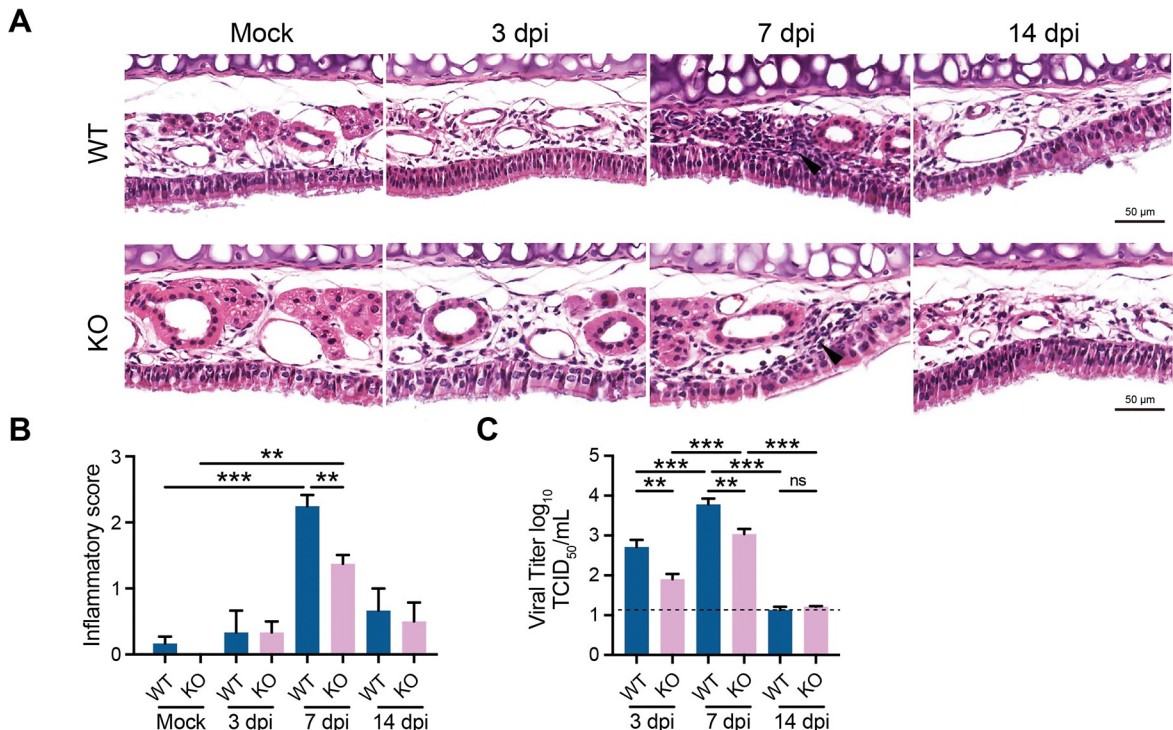

**Fig 2. IL-17RA exacerbates NP inflammation and tissue pathology during IAV.** WT C57BL/6 and IL17RA$^{-/-}$ (KO) mice were i.n. inoculated with 10 μl PBS containing 50xTCID$_{50}$ of PR8. Mice were euthanized at 3, 7, and 14 dpi for histological analysis of nasopharyngeal inflammation. **A-B.** H&E stain of mouse nasal septa and inflammatory scores. Black arrow: leukocytic infiltration. Scale bar: 50 μm. **C.** Viral titer present in nasal lavage determined by TCID$_{50}$. Data were representative of one experiment (n = 3) and expressed as SEM. Data were analyzed by One-Way ANOVA with Tukey's post-hoc in **B** and **C**. *p<0.05, **p<0.01, ***p<0.001.

extracellular trap formation, were significantly up-regulated in WT PR8 vs. WT Mock comparison. In contrast, the above inflammation-associated pathways remained un-detected in KO PR8 vs. KO Mock comparison. The Chemokine signaling pathway, C−type lectin receptor signaling pathway and Antigen processing and presentation were significantly up-regulated in both WT PR8 vs. WT Mock and PR8 vs. KO Mock comparison.

Next, we compared the significant pathways associated with signal transduction, inflammation, and cellular apoptosis. Our data show that pathways related to inflammation and cell death, such as apoptosis, JAK-STAT, and the TNF signaling pathways, were uniquely up-regulated in WT PR8 vs. WT Mock comparison (Fig 3D). These pathways remained un-detected in KO PR8 vs. KO Mock comparison. Since the majority of pathways belonging to the immune system were up-regulated in WT PR8 vs. WT Mock comparison, we presented the expression profiles for the DEGs belonging to Toll-like receptor signaling pathway, RIG-I-like receptor signaling pathway, NOD-like receptor signaling pathway, and Neutrophil extracellular trap formation pathways in the heatmap (Fig 3E). The genes such as *Irf7*, *Irf9*, *Stat2*, *Stat1*, and *Ccl4* were highly up-regulated in WT PR8 mice compared with WT Mock mice.

To validate RNAseq data, we performed magnetic purification of NP epithelial cells from WT and KO mice at 7 dpi. Based on the information from RNAseq, we performed western blot against significantly changing cellular markers between WT and KO mice. Our data show that compared to KO, the NP epithelial cells from WT mice had higher protein expression of several key inflammatory mediators, such as pSTAT1, SOCS1, SOCS3, IRF1, and IRF2 (Fig 3F). These data support RNAseq findings that epithelial cells from KO mice had reduced

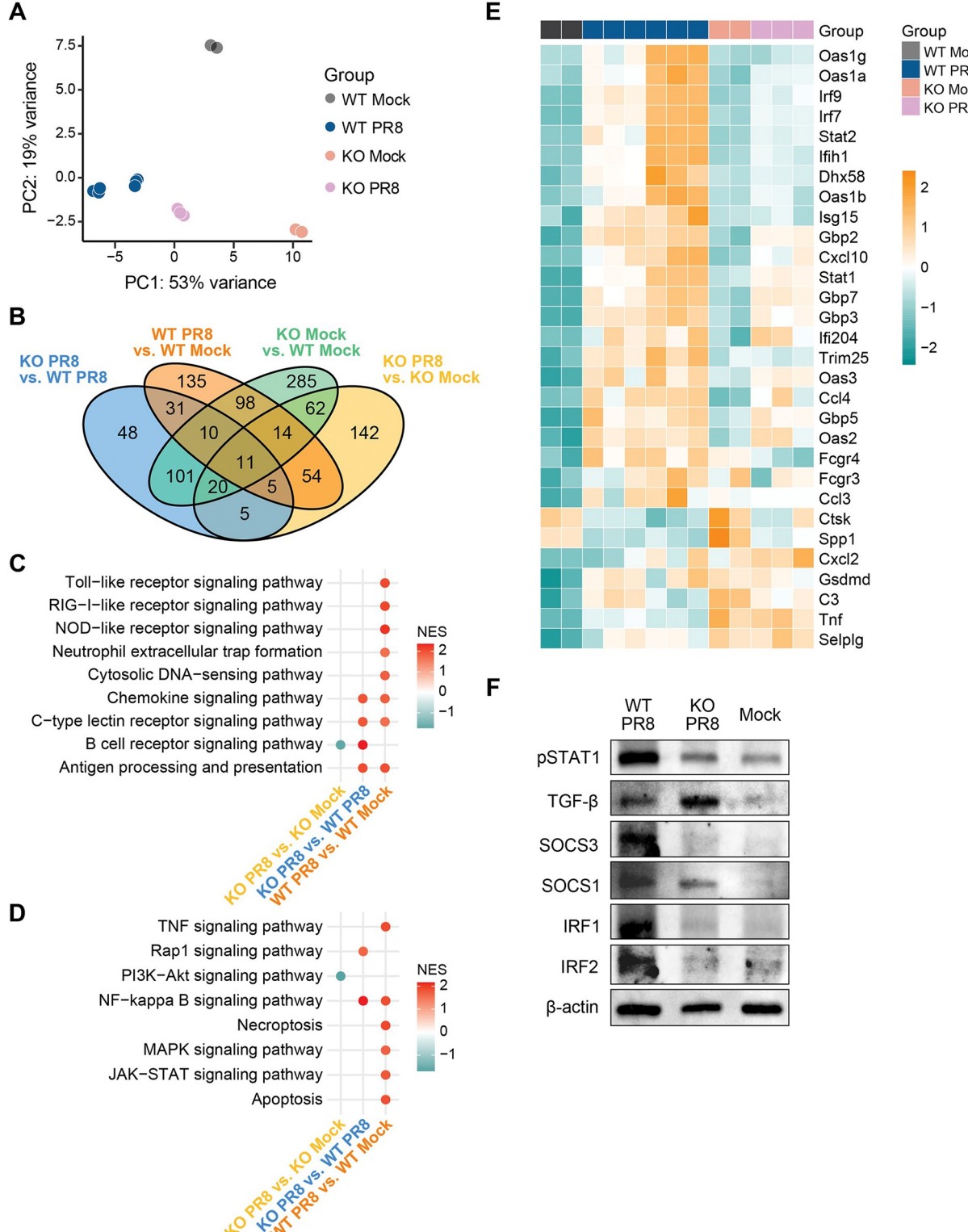

**Fig 3. Transcriptomic analysis of NP epithelail cells from WT and IL-17RA[-/-] mice.** WT C57BL/6 and IL17RA[-/-] (KO) mice were i.n. inoculated with 10 μl PBS containing 50xTCID$_{50}$ of PR8. Mice were euthanized at 7 dpi and NP epithelail cells were magnetically sorted for RNAseq. **A**. Principal component analysis (PCA) showed the group separation. Color stands for the groups. **B**. Venn Diagram showing the overlapping and unique differential expressed genes (DEGs) from all four comparisons. DEGs were determined by adjusted p-value <0.05 and an absolute log2 fold change >1. **C**. Dotplot showing the significant immune system pathways identified from Gene Set Enrichment Analysis

(GSEA) with KEGG pathway between all comparisons. NES stands for the normalized enrichment score of the significant pathway. The color stands for up-regulated (red) or down-regulated (blue) pathways. **D**. Dotplot showing the significant signal transduction and cell death associate pathways identified from Gene Set Enrichment Analysis (GSEA) with KEGG pathway between all comparisons. The color stands for up-regulated (red) or down-regulated (blue) pathways. **E**. Heatmap showing the expression profiling of DEGs belonging to significant pathways (Toll–like receptor signaling pathway, RIG–I–like receptor signaling pathway, NOD–like receptor signaling pathway and Neutrophil extracellular trap formation) between WT PR8 vs. WT Mock. **F**. Western blot analysis of NP epithelial cells from mock and IAV-infecetd WT and KO mice.

inflammatory responses. Overall, these data suggest that key molecular pathways regulating epithelial inflammation and cell death remained undetected in KO mice (up-regulated in WT mice), correlating with reduced NP inflammation and epithelial injury.

## IL-17RA$^{-/-}$ mice display increased resistance against *Spn-IAV* co-infection

Epithelial damage is a standalone risk factor of IAV-infected tissues for increased bacterial colonization and dissemination in the NP [21]. Since KO mice had reduced NP inflammation and epithelial injury, we postulated KO mice offer greater resistance against *Spn* co-infection. To address it, we utilized our previously published *Spn*-IAV co-infection model [12,22], where WT and KO mice were first colonized with *Spn* and later (24 h after *Spn* colonization) co-infected with IAV, mimicking the natural pathogenesis of *Spn*. *Spn* and IAV infections were performed using a low-volume (10 μL) intranasal inoculum typically used in colonization models. The co-infected mice were euthanized 6 days after establishing co-infection, and bacterial burden was determined in NP lavage and blood. In a parallel experiment, WT and KO co-infected mice were monitored to record survival for up to 2 weeks. Compared to WT, KO had reduced bacterial burden in the NP and the blood, suggesting reduced NP colonization and bacterial invasion in KO co-infected mice (Fig 4A). The reduced bacterial burden correlated with higher survival of KO mice (Fig 4B). Because WT and KO mice were co-infected with IAV 24 hours after *Spn* colonization, we compared *Spn* colonization rates between WT and KO mice after 24 hours of *Spn* colonization in the absence of IAV. Our data show that both WT and KO mice had similar *Spn* burden in NP lavage (Fig 4C), suggesting that differences in *Spn* bacterial density in the NP of co-infected WT and KO mice are indeed due to IAV-induced changes as opposed to early *Spn* control in the NP. These data suggest that IL-17RA is a significant pathologic axis causing upper respiratory epithelial injury and promoting invasive *Spn* infection during IAV co-infection.

## IL-17A promotes epithelial inflammation, cell death, and disrupts barrier function

Our *in vivo* data demonstrate the pathologic contribution of IL-17RA, including exacerbated epithelial injury and heightened expression of inflammatory and apoptotic pathways in NP epithelial cells from IAV-infected WT mice (compared to KO). Nonetheless, it remains to be established if IL-17RA exerts a direct pathologic effect on epithelial cells or if the reduced epithelial injury in KO mice is a consequence of reduced inflammatory response in the NP. To address this crucial question, we cultured primary human HNEpCs in ALI condition and investigated the effect of IL-17A treatment on epithelial injury and barrier function.

The HNEpCs were first expanded in submerged condition, and then 3.3x10$^4$ cells were seeded in each transwell and allowed to culture under ALI condition for up to 21 days. Epithelial differentiation was determined based on the expression of differentiation markers, i.e., α-acylated α-tubulin (cilia) and Muc5AC [23]. Compared to 7 days culture, a robust expression of α-acylated α-tubulin and Muc5AC was detected 21 days after ALI culture (Fig 5A–5C), confirming the differentiated epithelial cells. The differentiated HNEpCs (D21 ALI culture) were

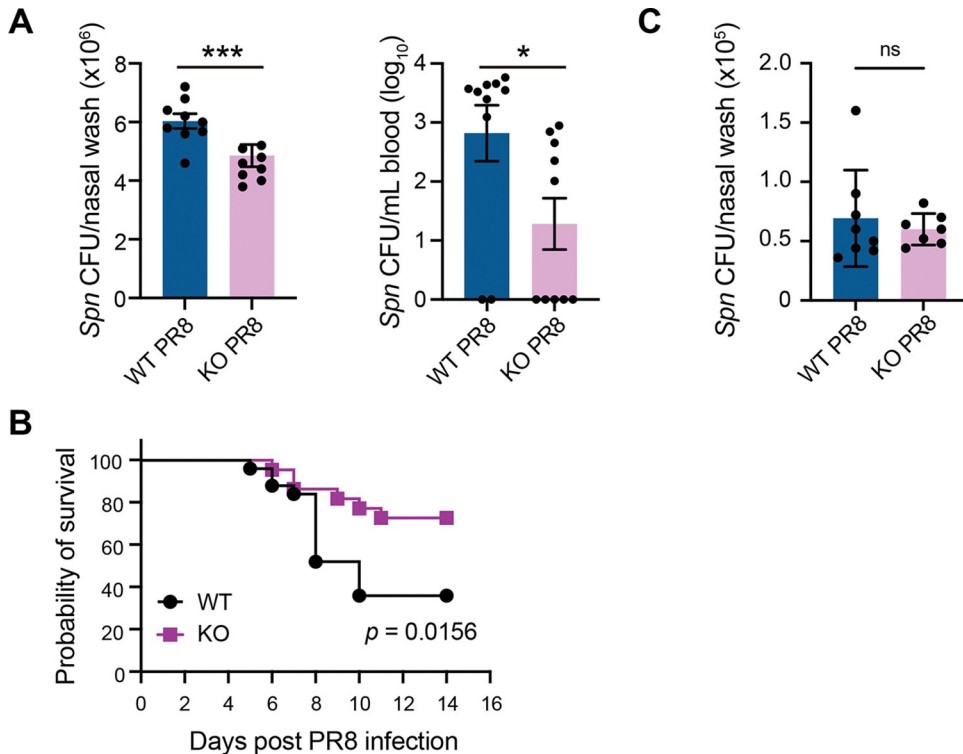

**Fig 4. IL-17RA deficiency restores protection against lethal *Spn* 6A-PR8 coinfection in murine nasopharynx.** WT C57BL/6 and IL17RA$^{-/-}$ (KO) mice were i.n. inoculated with 1x10$^5$ CFU of *Spn* 6A (BG7322) in 10 μL volume at day 0 and 10 μL of PR8 (50xTCID$_{50}$) on day 1 to establish co-infection. **A.** Mice were euthanized at 6 days post co-infection for bacterial burden calculation in the nasal lavage (left) and blood [46]. Data were calculated as number of colony forming units (CFU) of *Spn* in nasal lavage and CFU/mL of blood (Data were log10 transformed, and values equal to 0 were regarded as undetectable). Data shown were from one independent experiment (n = 8–10) and expressed as SEM. Data were analyzed by Student's t-test. *p<0.05, ***p<0.001. **B.** Mice were observed twice daily and monitored for survival. Data were representative of 2 independent experiments (n = 8/experiment) and analyzed using the Mantel-Cox log-rank test. **C.** WT C57BL/6 and IL17RA$^{-/-}$ (KO) mice were i.n. inoculated with 1x10$^5$ CFU of *Spn* 6A (BG7322) in 10 μL volume at day 0. Mice were euthanized 24 hours after *Spn* colonization and bacterial CFUs were determined in nasal lavage.

infected with human IAV (H1N1) pdm09 strain (MOI: 1). Mock or IAV-infected HNEpCs were supplemented with recombinant IL-17A (rIL-17A) for 24 hours. Epithelial cell death and barrier function were determined using a combined metric of immunofluorescence staining against annexin-V and ZO-1, basolateral transfer of FITC-dextran from apical to the basal transwell chamber, and *Spn* translocation from apical to basal transwell. IAV infection or rIL-17A treatment alone resulted in significant epithelial apoptosis, and IL-17A treatment caused a further increase in Annexin-V$^+$ HNEpCs (Fig 5D–5E). Since ZO-1, along with other TJ proteins, is crucial to regulating epithelial barrier function, we measured the expression of ZO-1 in IAV/rIL-17A-treated HNEpCs. Compared to mock infection, both IAV infection and rIL-17A treatment significantly decreased ZO-1 expression (Fig 5F) and increased the basolateral transfer of FITC-dextran to the basal transwell chamber (Fig 5G). However, rIL-17A treatment of IAV-infected epithelial cells produced a more significant decrease in ZO-1 expression (Fig 5F), which correlated with synergistically enhanced leakage of FITC-dextran to the basal transwell chamber (Fig 5G).

Next, we performed a bacterial translocation assay by infecting mock or IAV/rIL-17A treated HNEpCs with *Spn* in the apical chamber and measured bacterial recovery from the

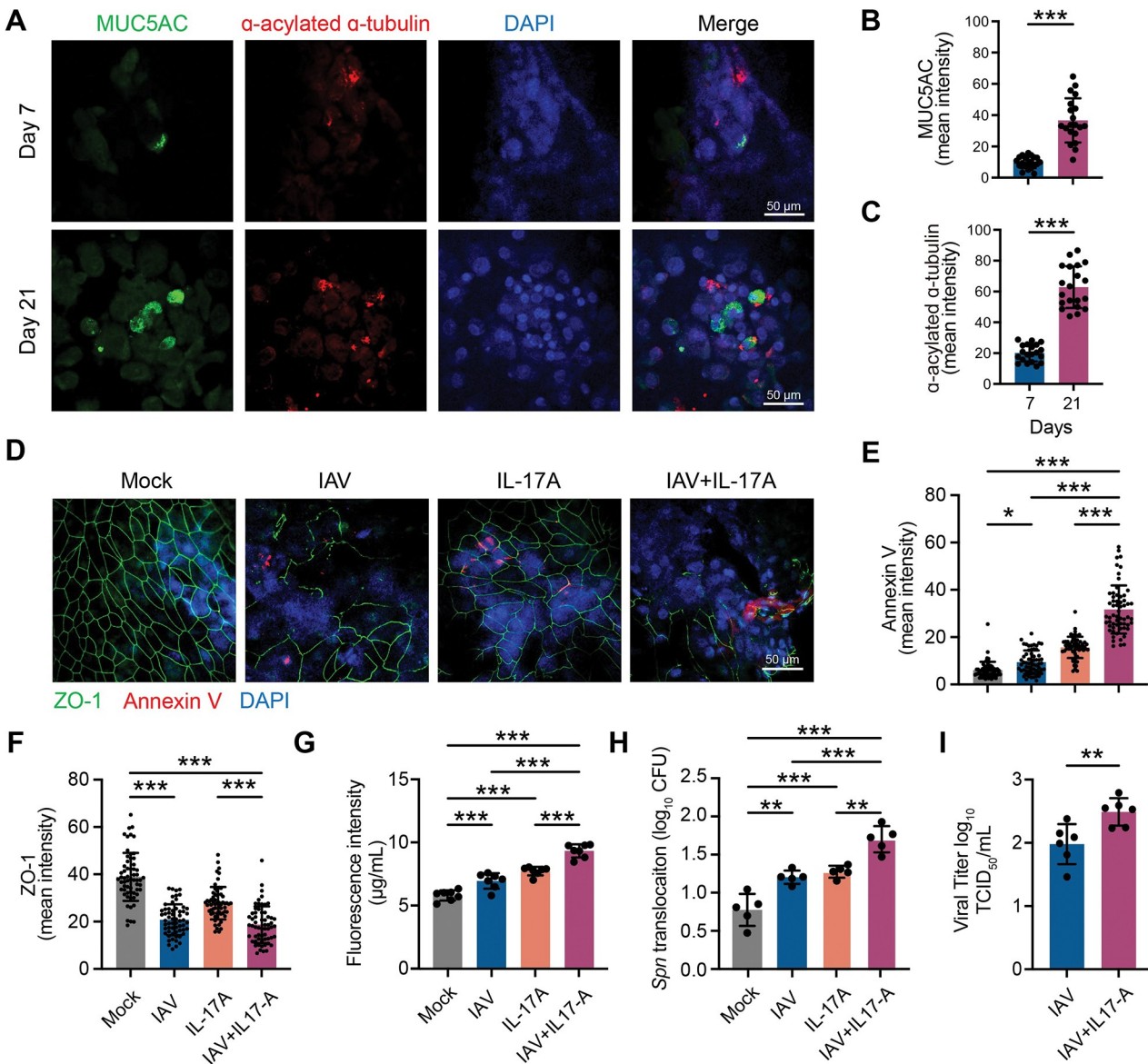

**Fig 5. IL-17A exacerbates IAV-mediated cell death and barrier disruption in air-liquid interface model of Human Nasal Epithelial cells (HNEpC). A-C.** HNEpCs differentiation in ALI culture was confirmed with anti-MUC5AC and anti-α-acetyltubulin immunofluorescence staining as described in the method section (**A**), and corresponding mean fluorescence intensity of MUC5AC and α-tubulin was quantified (**B&C**) with ImageJ software. Data were representative of 2 independent experiments (n = 10/experiment). **D-I.** Day 21 differentiated HNEpCs ALI cells were infected with Human IAV strain (H1N1) pdm09 strain A/California/04/2009 and subsequently cultured in the presence or absence of rIL-17A (IAV, IAV + rIL-17A) and incubated for 24 hours at 37°C in a humidified CO2 incubator. After incubation, the supernatant was collected for $TCID_{50}$, and the transwells were processed for immunofluorescence staining against Annexin-V and ZO-1 (**D-F**). Data were representative of 3 independent experiments (n = 20/experiment). **E-F.** Anti-Annexin-V and anti-ZO-1 mean fluorescence intensity quantification. **G**. HNEpCs monolayer permeability measurement through 4 kDa FITC-dextran, as described in the method section. **H.** HNEpCs monolayer permeability and translocation of *Spn* 6A (BG7322) bacteria from upper to lower transwell chamber. **I.** IAV viral load in transwell supernatants was quantified using standard $TCID_{50}$ assay. Data (**B, C,** and **I**) were analyzed by Student's t-test and Data (**E-H**) were analyzed by one-way ANOVA with Tukey's post-hoc. Data (**G-I**) were representative of 2 independent experiments (n = 3-4/experiment). *p<0.05, **p<0.01, ***p<0.001. Scale bar (**A,D**): 50 μm.

basal chamber of the transwell. The HNEpCs were infected with IAV or treated with rIL-17A as described for the FITC-dextran experiment and subsequently infected with *Spn* 6A (BG7322) for 3 hours in the MOI of 1. While IAV infection or rIL-17A treatment alone increased *Spn* translocation based on the recovered CFUs from basal transwell media, rIL-17A

treatment of IAV-infected HNEpCs significantly enhanced bacterial translocation as a higher number of *Spn* CFUs were recovered from the basal chamber of transwell media. Further, rIL-17A treatment enhanced viral spread in IAV-infected HNEpCs (Fig 5I), supporting *in vivo* data (Fig 2C). These data broadly support the FITC-dextran experiment that rIL-17A treatment of IAV-infected epithelium overwhelmingly disrupts the barrier integrity, resulting in leaky epithelia that facilitate *Spn* translocation across damaged epithelial monolayer.

## Discussion

In this study, we investigated the role of IL-17RA as a contributor to IAV host defense and tissue damage in the upper respiratory tract. The major findings of our study are that during IAV infection, IL-17RA is a significant regulator of neutrophilia, promotes NP inflammation, tissue pathology, and converts *Spn* bacterial colonization to an invasive infection. The NP epithelial cells from IL-17RA$^{-/-}$ mice exhibited ameliorated epithelial injury, correlating with attenuated expression of cellular pathways regulating inflammation and cell death. Notably, IL-17RA deficiency mitigated NP inflammation and suppressed viral spread. In an ALI epithelial model using primary human NP epithelial cells, rIL-17A treatment promoted epithelial cell death and disrupted the barrier integrity, supporting the *in vivo* data. Thus, our data reveal the pathologic contribution of IL-17RA during IAV upper respiratory tract infection.

The IL-17RA axis is broadly protective against mucosal pathogens, including respiratory bacteria, as IL-17-deficient mice fail to clear bacterial colonization in the upper respiratory tract [8,9,24,25]. IL-17RA mediates protection by regulating neutrophil recruitment crucial to bacterial control at mucosal surfaces [13,14]. Paradoxically, the pathologic role of IL-17RA has been previously shown in influenza lung models, as IL-17RA deficient mice had reduced lung pathology and higher survival [10]. Similarly, intranasal treatment of IAV-infected mice with IL-17 inducible mucosal adjuvants exacerbated lung pathology resulting in reduced survival [11]. We recently showed the double-edged function of IL-17A; while protective against *Spn* colonization, IAV-induced IL-17A converted commensal *Spn* colonization into an invasive bacterial infection [12]. Further, a recent report suggested that while augmenting IL-17 responses against pneumococci may decrease nasal colonization, it may worsen the outcome during pneumonia caused by some serotypes [26]. Consistent with the above findings, our data show that KO mice had reduced CXCL1 and neutrophil recruitment, which correlated with reduced epithelial inflammation, tissue pathology, and reduced permisvenss of KO mice to *Spn* co-infection at 7 dpi. However, both groups (WT and KO) effectively resolved NP tissue inflammation and viral load by 14 dpi, suggesting that IL-17RA induced pathologic effect is only limited during the peak inflammation (7 dpi), and IL-17RA does not produce long-term defects in the resolution of inflammation or IAV viral control in WT mice.

As both bacteria and IAV can activate IL-17RA leading to differential outcomes *vis-à-vis* the resolution of infection and tissue damage, the notable difference between the two is potentially the magnitude of IL-17RA activation. While bacterial colonization elicits a mild Th17 response that promotes neutrophil recruitment and gradual clearance of colonization in the NP [9], IAV-induced IL-17A triggers overwhelming neutrophil recruitment [12], which can cause significant injury to epithelial cells in the NP and can promote bacterial colonization, outgrowth, and dissemination. These observations are supported by undetectable IL-17A at the protein level in the NP lavage of *Spn*-colonized mice [12]. Instead, IL-17A-mediated clearance of *Spn* colonization correlates with the presence of antigen-specific CD4$^+$ T cells (Th17) in lung or spleen. On the contrary, IL-17A is robustly detected at the protein level in NP lavage of IAV-infected mice, correlating with more exuberant neutrophil recruitment in the NP. No differences in macrophages and monocytes were observed between WT and KO mice,

suggesting the dispensability of IL-17RA axis in regulating non-neutrophilic responses during IAV infection in the upper respiratory tract.

The balanced IL-6 and TGF-β responses are crucial to Th17 polarization and IL-17A response generation [27]. In addition, TGF-β also plays a crucial role in promoting tissue repair [28]. While WT and KO mice had some noticeable differences in IL-6 and TGF-β protein levels in nasal lavage, KO mice did not exhibit any defects in IL-17A production, as similar IL-17A protein levels were found in NP lavage of WT and KO mice. While IL-17RA promotes neutrophilic inflammation [29,30], little is known about its role in regulating epithelial inflammation, the primary site of viral-bacterial interactions in the NP. The findings from gut mucosal models suggest the double-edged role of IL-17A on epithelial barrier [18]. The IL-17RA can promote epithelial inflammation by regulating the CXCL1/2-dependent neutrophil recruitment [13,14], which can establish a direct cross-talk with epithelial cells. Further, IL-17A can exert a direct pathologic effect on epithelial cells by promoting cellular apoptosis and disrupting barrier integrity, as shown in gut models [17,31,32]. To address this, we first performed RNAseq on NP epithelial cells from IAV-infected WT and KO mice. Our RNAseq findings reveal that epithelial cells from WT mice presented several key pathways regulating inflammation and cell death, which were missing in KO epithelial cells. In particular, we observed that pathways associated with inflammation and cell death, such as apoptosis, JAK-STAT, and the TNF signaling pathways, were uniquely presented in WT mice, and lacked in IL-17RA KO mice. Interestingly, KO mice had higher TGF-β protein levels in NP lavage, which correlated with reduced inflammation and epithelial injury in KO mice. Further, the higher TGF-β protein expression in epithelial cells from KO mice suggests that NP epithelial cells are a significant source of TGF-β, which can repair damaged epithelial cells in the NP. However, TGF-β responses in the NP are likely derived from both epithelial and immune cells, which deserve further investigation. Further, it remains to be determined if IL-17RA directly suppresses epithelial TGF-β response or if a higher TGF-β protein level results from reduced inflammatory response in the NP of KO mice. These findings are the subject of ongoing investigations in the lab.

IAV-induced epithelial inflammation in the NP is a significant risk for increased bacterial colonization and transmission [33,34]. The reduced bacterial burden (NP and blood) in IL-17RA KO mice suggests that IL-17RA signaling is sufficient to promote epithelial injury in the NP, allowing *Spn* colonization to develop into an invasive infection during IAV. To validate the *in vivo* findings and investigate whether IL-17A can mediate pathologic effects on epithelial barrier integrity, we grew primary human NP epithelial cells in ALI condition. We determined the impact of IAV infection and rIL-17A treatment on epithelial inflammation and barrier function. The ALI epithelial model more closely mimics the *in vivo* physiology of respiratory epithelial cells [35,36]. While rIL-17A treatment promoted epithelial cell death and enhanced epithelial permeability, IAV infection and rIL-17 treatment resulted in a synergistic increase in epithelial inflammation, cell death, and permeability. These findings indicate that while IL-17A treatment itself induced cell death and had a deleterious effect on epithelial barrier function, it can synergistically enhance epithelial cell death and disrupt the barrier integrity of IAV-infected epithelial cells.

In summary, our findings unveiled the pathologic role of IL-17RA during upper respiratory IAV infection. Given that the NP niche is a primary site of bacterial-viral interactions, our results suggest that IL-17RA-mediated epithelial injury can allow commensal bacterial colonization to develop in invasive infection during *Spn*-IAV co-infection. Our findings suggest that IL-17RA is a significant pathologic axis that could be utilized as a therapeutic target to treat influenza upper respiratory infections. Future investigations are required to determine whether IL-17RA is a common pathologic framework against respiratory viruses other than influenza.

## Materials and methods

### Ethics statement

Wild-type (WT) C57BL/6 mice were purchased from Jackson Laboratory and bred in-house, following the guidelines of the University of North Dakota Animal Care and Use Committee. IL17RA$^{-/-}$ are the propriety of Amgen (Thousand Oaks, California) and were provided by Sarah Geffen (University of Pittsburgh, Pittsburg, PA). The animal work undertaken in this study was reviewed and approved by the University of North Dakota Institutional Animal Care and Use Committee (Protocol# 1907–2) and University of Florida Institutional Animal Care and Use Committee (Protocol# IACUC202100000057).

### Animals and microbial strains

Equal proportions of age-matched (6-8-week-old) male and female mice were included in the study. Mouse-adapted influenza A H1N1 A/Puerto Rico/8/1934 (IAV or PR8) virus was purchased from Charles River. Human IAV strain (H1N1) pdm09 strain A/California/04/2009 was purchased from American type culture collection (ATCC). *Spn* serotype 6A strain BG7322 was provided by Rochester General Hospital Research Institute and has been previously used by us and others [22,37,38].

### Infection models

Mice were anesthetized with isoflurane (4% v/v isoflurane/oxygen) and intranasally (i.n.) inoculated with 10 μL PBS containing 1x10$^5$ CFUs of *Spn* 6A (BG7322). To establish co-infection, *Spn* colonized mice were i.n. infected with PR8 (10 μL volume containing 50xTCID$_{50}$). At 6 days post co-infection, mice were euthanized, and retrograde NP lavage fluid was collected using 200 μL of sterile PBS. Blood was collected *via* cardiac puncture using a sterile 1.0 mL syringe. NP lavage and blood samples were serially diluted and plated on blood agar, and CFUs were enumerated the next day. To compare the differences in *Spn* colonization between WT and IL-17RA$^{-/-}$ mice, mice were anesthetized and i.n. inoculated with 10 μL PBS containing 1x10$^5$ CFUs of *Spn* 6A (BG7322). Mice were euthanized 24 hours after infection, and retrograde NP lavage fluid was collected using 200 μL of sterile PBS. NP lavage and blood samples were serially diluted and plated on blood agar, and CFUs were enumerated the next day.

### Flow cytometry

Blood samples were prepared for flow cytometry as previously described [39]. Briefly, RBCs were treated with ammonium-potassium-chloride (ACK) lysis buffer (Life Technologies) for 10 min on ice and washed with PBS containing 5% FBS. Lavage samples were first pooled and 100 μl of diluted Fc block was added to each lavage and blood sample and incubated for 20 min on ice. The cells were centrifuged and supernatants were discarded. Blood and lavage samples were subsequently stained for antibodies against CD45 (BV421), Cd11b (APC-Cy7), Ly6C (BV711), and Ly6G (PE) for 30 min at room temperature in dark, as described earlier [12,40]. A BD FACSymphony A3 Flow Cytometer collected 100,000 events for blood samples and 50,000 events for lavage samples. The data was analyzed using FlowJo. For cytokine/chemokine analysis of NP lavage, a customized 14-plex (LEGENDplex bead-based immunoassays) was used per manufacturer's recommendation (BioLegend). Samples were acquired using a BD FACSymphony A3 Flow Cytometer and analyzed using LEGENDplex V8.0 Data Analysis Software (BioLegend).

## NP histology

Mock and IAV-infected WT and IL-17RA$^{-/-}$ mice were i.n. infected with 10 μL PBS containing 50xTCID$_{50}$ of IAV, and on 3, 7, and 14 dpi, the histology samples of the nasopharynx were prepared as previously described [37]. Briefly, the skin was removed from the skull, and the anterior nasopharynx was removed and placed in 10% neutral buffered formalin for 24 hours prior to being stored in 70% ethanol. Decalcification, paraffin embedding, sectioning, and H&E staining were performed at University of North Dakota and University of Florida histology cores. The H&E-stained slides were coded and presented to 2–3 blinded observers to score on a range of 0 (little to no infiltration below NP epithelium) to 4 (maximum cellular infiltration below NP epithelium). The histological images were acquired using a NanoZoomer 2.0-HT Brightfield Fluorescence Slide Scanning System (Hamamatsu Photonics, Japan) (University of North Dakota) and by Leica ScanScope CS (Leica, CA, USA) and analyzed using NDP.view2 Viewing Software (Hamamatsu).

## RNA sequencing

WT and IL-17RA$^{-/-}$ mice were i.n. infected with 10 μL PBS containing 50xTCID$_{50}$ of IAV. Mice were euthanized at 7 dpi, and NP epithelial cells were magnetically sorted using CD45 and Epcam beads (CD45$^-$ Epcam$^+$). RNA was extracted using a Qiagen RNA extraction kit, and the sequencing was performed at Genomic Core of the University of North Dakota. Preliminary processing of raw reads was performed using Casava 1.8 (Illumina, Inc., San Diego, CA). Subsequently, the overall QC metrics were obtained using FastQC (https://www.bioinformatics. babraham.ac.uk/projects/fastqc/). Low-quality reads and those containing poly-N were removed to ensure high-quality mapping. The clean reads were mapped using the mouse genome mm10 as a reference with HISAT2 [41]. The number of reads aligning to genes was counted with featureCount [42]. DESeq2 [43] was used to identify significant changes in gene expression between groups, and P values were corrected for multiple testing using the Benjamin-Hochberg method. The genes with adjusted p-values <0.05 and |log2 fold change| >1 were identified as the significant differentially expressed genes (DEGs). Gene set enrichment analysis (GSEA) with Kyoto Encyclopedia of Genes and Genomes (KEGG) pathways was performed with our in-house R package (https://github.com/hurlab/richR). The Benjamin-Hochberg corrected p-value of 0.01 was used to choose the significantly enriched pathways.

## Expansion and differentiation of HNEpCs under air-liquid interface conditions

Primary Human Nasal Epithelial Cells (HNEpCs) were purchased from Promo Cell (Cat# C-12620). The cells were revived and maintained as per the manufacturer's instructions. Briefly, in the expansion phase, approximately 33,000 cells were seeded per trans-well (Corning, Cat# 3470) and maintained under submerged conditions using was Promo cell growth medium. After 3–4 days, when confluent monolayer formed, the growth media was replaced with ALI maintenance media (Stem cell technologies, PneumaCult-ALI Medium Cat# 05001; Heparin Solution, Cat# 07980; Hydrocortisone, Cat# 07925) the basal chamber leaving the apical chamber empty. Basal chamber media were changed every two days. ALI culture was maintained for up to 21 days for mucociliary kinetic studies. Epithelial differentiation was established by performing immunofluorescence staining against MUC5AC and anti-α-acetyltubulin (cilia). In brief, the transwell insert was fixed in ice-cold (-20˚C) methanol and incubated overnight at -20˚C followed by cold acetone (-20˚C) fixation for 1 minute. The transwell membrane was blocked by blocking solution at room temperature for 1 hour. Primary antibodies (Anti-

MUC5AC, Cat# 61193, cell signal technology; Anti-Acetyl-α-Tubulin, Abcam, Cat# ab24610) were added on the apical side, incubated at 4˚C overnight, followed by incubation with secondary antibody for 3 hours at 4˚C in the dark. The nuclear staining was performed using DAPI. The images were captured using a Zeiss 710 confocal laser scanning microscope at Center for Immunology and Transplantation, University of Florida. The image processing and quantification were done with NIH Fiji (ImageJ) software.

## IAV infection, IL-17A treatment, and Analysis of epithelial barrier function

At day 21 of ALI culture, the monolayer of differentiated epitehlial cells was washed with DPBS (stem cell technologies), and the cells were incubated with A Human IAV strain (H1N1) pdm09 strain (MOI:1) for 2 hours in 100 μl basal growth medium (Promo cell Cat# C-21260). The cells were washed twice with DPBS and further stimulated with human recombinant IL-17A (rIL-17A: 200 ng/mL) in the presence or absence of IAV, in ALI maintenance media (stem cell PneumaCult-ALI Medium, Cat# 05001) for 24 hours at 37˚C in $CO_2$ incubator. The epithelial barrier function was analyzed based on immunofluorescence (IF) staining (confocal) based expression of Annexin-V (Cat# 66245-1-IG, Protein tech), tight junction protein ZO-1 (Cat# 61–7300, ThermoFisher Scientific), measuring the basolateral transfer of 4kDa Fluorescein Isothiocyanate-Dextran (FITC-Dextran, Cat# 4013, Chondrex Inc.) from apical to basal chamber of the transwell, and *Spn* bacterial translocation across damaged epithelium to the basal chamber of transwell. The IF staining was performed as described above, and image processing and quantification were done with NIH Fiji (ImageJ) software. To assess the membrane integrity of epithelial monolayer, we measured the basolateral transfer of FITC-Dextran. Briefly, 200 μL (3 mg/mL) of FITC-dextran was added to the apical chamber of the transwell, and 500 μL of ALI maintenance media was added in the basal chamber and incubated at 37˚C in a $CO_2$ incubator for 2 hours. After incubation, 50 μL of media from the basal chamber was collected, serially diluted, and transferred to 100 μL of diluted standards and samples to a black 96-well plate and read on a fluorescence reader as per manufactures instructions (Excitation: 490 nm/Emission: 520 nm). *Spn* membrane translocation assay was performed as described earlier [44]. Briefly, *Spn* 6A (strain BG7322) was washed and resuspended in antibiotic-free DMEM media and added to apical chamber of the transwell in the MOI of 1 and incubated at 37˚C in a $CO_2$ incubator for 3 hours. After incubation, the culture medium in the lower transwell chamber was collected, serially diluted, and plated on blood agar plates. The bacterial colnies were enuemarted the next day.

## Quantitation of IAV load

The viral load in transwell supernatants or NP lavage fluids were quantified using $TCID_{50}$ assay [45]. In brief, a confluent monolayer of MDCK cells (96 wells) was incubated with 10-fold serial diluted lavage or cell supernatants in infection media (DMEM, 0.3% BSA, 1% Pen/strep, 1 μg/ml TPCK-Trypsin) and the plate was incubated at 37˚C in $CO_2$ incubator for 5 days. After 5 days post-infection, the MDCK cells were fixed and stained with 0.5% crystal violet solution. Each well of a plate was scored against negative control (non-infected media only) for viral growth. The $TCID_{50}$ values were calculated following the published protocol [45].

## Cytokine analysis

The quantitation of cytokines was performed using a customized 14 plex LEGENDplex bead-based multi-analyte detection kit (Biolegend, San Diego, USA), following manufacturer's instructions. Data were analyzed using the LEGENDplex Data Analysis Software Suite.

## Western blot

WT and IL-17RA$^{-/-}$ mice were i.n. infected with 10 μL PBS containing 50xTCID$_{50}$ of IAV. Mice were euthanized at 7 dpi, and NP epithelial cells were magnetically sorted using CD45 and Epcam beads (CD45$^-$ Epcam$^+$). Cells were homogenized in T-PER lysis buffer with a protease and phosphatase inhibitor mixture (ThermoFisherScientific). Samples were centrifuged at 14,000 x g for 10 min at 4°C, and supernatants were aliquoted and preserved at -80°C. Twenty micro grams of protein lysates were used for electrophoresis. Western blotting was conducted as previously described [17].

## Acknowledgments

We thank Donna Laturnus (UND) and Jinwan Yang and Dongtao Fu (UF) for their help with histology.

## Author Contributions

**Conceptualization:** Nadeem Khan.

**Data curation:** Zahrasadat Navaeiseddighi, Jitendra Kumar Tripathi, Kai Guo, Zhihan Wang, Taylor Schmit, Delano R. Brooks, Reese A. Allen, Nadeem Khan.

**Formal analysis:** Zahrasadat Navaeiseddighi, Jitendra Kumar Tripathi, Kai Guo, Zhihan Wang, Taylor Schmit, Nadeem Khan.

**Funding acquisition:** Nadeem Khan.

**Investigation:** Kai Guo, Nadeem Khan.

**Methodology:** Jitendra Kumar Tripathi, Kai Guo, Zhihan Wang, Taylor Schmit, Reese A. Allen, Nadeem Khan.

**Project administration:** Nadeem Khan.

**Resources:** Nadeem Khan.

**Software:** Kai Guo, Zhihan Wang, Taylor Schmit.

**Supervision:** Nadeem Khan.

**Validation:** Zahrasadat Navaeiseddighi, Zhihan Wang, Taylor Schmit, Delano R. Brooks, Reese A. Allen.

**Visualization:** Zahrasadat Navaeiseddighi, Jitendra Kumar Tripathi, Kai Guo, Zhihan Wang, Taylor Schmit.

**Writing – original draft:** Zahrasadat Navaeiseddighi, Jitendra Kumar Tripathi, Kai Guo, Zhihan Wang, Taylor Schmit, Junguk Hur, Ramkumar Mathur, Donald Jurivich, Nadeem Khan.

**Writing – review & editing:** Zahrasadat Navaeiseddighi, Jitendra Kumar Tripathi, Kai Guo, Zhihan Wang, Taylor Schmit, Reese A. Allen, Junguk Hur, Ramkumar Mathur, Donald Jurivich, Nadeem Khan.

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
