## [Decision Letter · Decision Letter 0]

17 Apr 2023

Dear Professor Khan,

Thank you very much for submitting your manuscript "IL-17RA promotes pathologic epithelial inflammation in a mouse model of upper respiratory influenza infection" for consideration at PLOS Pathogens. As with all papers reviewed by the journal, your manuscript was reviewed by members of the editorial board and by several independent reviewers. In light of the reviews (below this email), we would like to invite the resubmission of a significantly-revised version that takes into account the reviewers' comments.

In particular, the following points need to be addressed experimentally: 

- Timing of co-infection: Does prior administration of Spn to WT and KO mice leads to the same level of colonization and inflammation, as raised by reviewer 1? 

- The use of ALI at day 7, which contain no ciliated cells, and appear leaky, is unusual and will make it impossible to place the findings of this study in the body of existing data which use mature ALI cultures. It is critical that experiments with ALI are performed with fully mature ALI (day 21), and, preferably, using a human IAV, as pointed out by reviewer 2. 

- Determine the kinetics of the IAV infection/inflammation in the NP with one earlier and one later time point in both WT and KO mice, as raised by reviewer 3. 

We cannot make any decision about publication until we have seen the revised manuscript and your response to the reviewers' comments. Your revised manuscript is also likely to be sent to reviewers for further evaluation.

Sincerely,

Amy L. Hartman, PhD

Academic Editor

PLOS Pathogens

Meike Dittmann

Section Editor

PLOS Pathogens

Kasturi Haldar

Editor-in-Chief

PLOS Pathogens

orcid.org/0000-0001-5065-158X

Michael Malim

Editor-in-Chief

PLOS Pathogens

orcid.org/0000-0002-7699-2064

Reviewer's Responses to Questions

**Part I - Summary**

Reviewer #1: In this manuscript, Tripathi et al. interrogated the role of the interleukin-17A (IL-17A) and IL-17 receptor (IL-17RA) signaling in host-defense against two human pathogens, influenza A virus (IAV) and the bacterium Streptococcus pneumoniae. The authors combined in vitro and in vivo infection models to study how loss of IL-17RA impacts the immune response and tissue damage in the upper respiratory tract upon the infection with IAV. The authors demonstrated that IL-17RA knockout mice have decreased recruitment of neutrophils, inflammation, and tissue pathology upon viral infection. They also found that IL-17RA knockout mice were more resistant to S. pneumoniae during coinfection with IAV than WT mice. Finally, the authors established that the addition of IL-17A to IAV infected epithelial cells increased inflammation and cellular pathology and disrupted barrier function. Overall, the manuscript is well-written, and the conclusions are justified by the data. There are a few comments for the author’s consideration.

Reviewer #2: In this study (PPATHOGENS-D-23-00456), Tripathi and colleagues examine the role of IL-17RA as a mediator of host defense and tissue damage in the upper respiratory tract during an influenza virus infection. They convincingly show, in the mouse model, that IL-17RA is a significant regulator of neutrophilia, promotes inflammation in the nasopharynx and promotes Spn colonization to invasive infection. Furthermore, the authors used an in vitro ALI model to examine the impact of IL-17A treatment although the relevance of this chosen model system with non-ciliated human cells and a mouse adapted IAV needs to be clarified. The paper is well written and comprehensible. The dual role of IL-17RA is very interesting and examining it in the mucosal site is very relevant to IAV infection. A better understanding of how influenza virus infection potentiates bacterial secondary infection is important as these are quite common.

Reviewer #3: The study from Tripathi et al. focuses on the role of IL-17RA in the context of IAV infection and coinfection with S. pneumoniae. This study is a follow-up work where they previously had shown the role of IL17 being important during Spn-IAV coinfection. Using a murine model of NP infection combined with transcriptomics and an in vitro cell culture model, they show that IL17RA signaling causes epithelial damage, neutrophil influx, increased viral titers, and increased Spn dissemination to distal sites. The study is well put together, but the reviewer has some concerns with the investigation.

**Part II – Major Issues: Key Experiments Required for Acceptance**

Reviewer #1: 1. In figure 3A the data presentation for Spn in the blood is a little confusing. Are the data points that seem to be sitting on the x-axis really zero or are they a substantially higher value that cannot be deciphered because of the scale chosen for the y-axis? Perhaps a log scale would be more appropriate or log-normalized data? This way the reader could better assess the data.

2. Have the authors checked to see that prior administration of Spn to WT and KO mice leads to the same level of colonization after 24 hours? Do KO mice just have less bacteria to begin with, prior to administration of IAV? This seems to be an important control that was not considered.

Reviewer #2: The authors state that the respiratory tract is primarily lined by pseudostratified ciliated cells. They show that at D21 the ALI cultures have more ciliated cells than D7, which would be more physiological. What is the rational for using D7, if D21 are more relevant? Also, the values of ZO-1 in mock at D7 in Fig 4E and 5C are quite different. Can the authors explain this?

In Fig 5E, is the amount of leakage seen in the mock sample normal? Would ALI cultures on D21 have less leakage because they have more ZO-1/tight junctions? Did the authors measure the TEER of their transwells?

What is the reasoning behind using PR8, a mouse adapted IAV, in the ALI model, which is composed of human NP epithelial cells? Fig 5F shows low levels of replication. Would using a human IAV that replicates better in human cells demonstrate better fold changes between infection, IL-17A treatment and the combination?

In the in vitro model, would the damage caused by infection or IL-17A treatment be significant enough to facilitate enhanced replication of Spn?

Reviewer #3: 1. Their murine model only uses a single time-point to study to look at the inflammation associated with IAV infection. As they point out, the role of IL17RA has not been examined in the context of IAV infection, and it would be pertinent to look at the kinetics of the IAV infection/inflammation in the NP with earlier and later time points in both WT and KO mice. Do the KO mice always have lower inflammation, or do they eventually catch up to the WT-infected mice?

2. Even though this reviewer found their RNA-seq study fascinating with interesting results, they almost feel like they are tacked on to the study, as there is no follow-up on the results they obtained. Transcript levels also might not correlate with protein levels in the NP. It would help if they looked downstream, for example, RIG-1 or the NF-kappa B signaling pathway, and looked at protein levels as they did in Figure 1.

3. The differences observed in membrane permeability are minimal in Figure 5E. They only show a one-time point at 2 hrs. Doing a kinetic run to determine how membrane permeability is impacted over time would be appropriate.

**Part III – Minor Issues: Editorial and Data Presentation Modifications**

Reviewer #1: 1. The figure callouts in the text for Figure 5 were not always clear. For example, the FITC dextran data is referred to several times, but was never called out as Fig 5E in the text. Consider being more thorough with figure callouts to make referring back more straightforward.

2. Is there a statistic missing from figure 5A (IL-1b)? There are no statistical comparisons shown for the PR8+IL17-A condition.

3. Did the authors block Fc receptors in their antibody staining for flow? The methods in general are a bit thin. It would be helpful to increase the level of detail throughout.

4. Line 90, should be 10ul instead of 10ml.

Reviewer #2: Please define what is IL-17RA at the beginning.

Can the authors provide some context for the reader as to the significance of looking at levels of IL-6, TNF-a and TGF-b?

Line 90 should probably read 10uL as the low volume.

In Fig 4, the authors are comparing D7 vs D21, which is hard since the y-axis are different in some panels. Please include on the same graph for easier comparison.

I would be helpful to the reader to include a model.

Reviewer #3: 1. In Figure 3, they show lower levels of S. pneumoniae levels in the NP (KO vs. WT) correlate with lower levels in the blood. They should normalize their levels of bacteria in the blood to what is present in the NP, as that would be a more accurate comparison.

2. The strain of Spn used is only mentioned in one of the figure legends. It would be appropriate to describe the strain in the method section and the reference if available.

3. In line 90, the volume for infection with IAV should be ul and not mL.

4. Treatment with IL17A should say recombinant.

5. In Figure 5, an extra hyphen is added to IL-17A with treatment with PR8.

PLOS authors have the option to publish the peer review history of their article (what does this mean?). If published, this will include your full peer review and any attached files.

Reviewer #1: No

Reviewer #2: No

Reviewer #3: No
---

## [Editor Report · Decision Letter 1]

20 Nov 2023

Dear Professor Khan,

We are pleased to inform you that your manuscript 'IL-17RA promotes pathologic epithelial inflammation in a mouse model of upper respiratory influenza infection' has been provisionally accepted for publication in PLOS Pathogens.

Best regards,

Amy L. Hartman, PhD

Academic Editor

PLOS Pathogens

Meike Dittmann

Section Editor

PLOS Pathogens

Kasturi Haldar

Editor-in-Chief

PLOS Pathogens

orcid.org/0000-0001-5065-158X

Michael Malim

Editor-in-Chief

PLOS Pathogens

orcid.org/0000-0002-7699-2064
---

## [Editor Report · Acceptance letter]

1 Dec 2023

Dear Professor Khan,

We are delighted to inform you that your manuscript, "IL-17RA promotes pathologic epithelial inflammation in a mouse model of upper respiratory influenza infection," has been formally accepted for publication in PLOS Pathogens.

Best regards,

Michael Malim

Editor-in-Chief

PLOS Pathogens

orcid.org/0000-0002-7699-2064